# Future Sequon Finder - A novel approach for predicting future N-linked glycosylation sequon locations on viral surface proteins

**Shane P. Bryan[1], Martin S. Zand** [iD][1,2]*

**1** Department of Medicine, Division of Nephrology, University of Rochester Medical Center, Rochester, New York, United States of America, **2** Clinical and Translational Science Institute, University of Rochester Medical Center, Rochester, New York, United States of America

* Martin_Zand@urmc.rochester.edu

**Data availability statement:** All relevant data are within the manuscript and its Supporting information files.

## Abstract

Influenza viruses are known to evade host immune responses by shielding vulnerable surface protein epitopes via N-linked glycosylation. A program titled *Future Sequon Finder* was developed to predict the locations in which glycan binding sites are most likely to emerge in future influenza hemagglutinin proteins. The predictive modeling approach considers how closely sites in currently circulating strains resemble glycosylation sequons at the nucleic acid level, the surface accessibility of those sites, and the mutation frequency of amino acids at those sites that would need to change to form a glycosylation sequon. The efficacy of this model is tested using historic human H1N1 and H3N2 influenza strains along with swine H1N1 strains. Through this analysis, it is revealed that glycosylation addition events in influenza hemagglutinin proteins are typically the result of single nucleotide mutation events. It is also demonstrated that site-specific mutation frequency and surface accessibility are powerful predictors of which sites will become glycosylated in human influenza viruses when considered with the genetic composition of the sites in question. Having been designed to incorporate these factors, the program successfully predicted almost every historic sequon addition event (28/30 in human IFVs, 14/15 in swine IFVs). For human strains, it also ranked the correct near-sequons highly among falsely predicted sequons based on site-specific mutation frequency. After demonstrating the model's power with historical data, the program was used to predict future HA glycosylation sequon locations based on currently circulating human influenza viruses.

## Introduction

Glycosylation of viral proteins that bind to cell surface ligands is a key evolutionary adaptation, as it alters protein antigenicity and shields target sites from immune responses [1–6]. Knowledge of actual or potential N-linked glycosylation sites is important for identifying epitopes on viral surface proteins that serve as ideal target sites for antibodies [7], vaccine engineering [8–10], or immune epitope focusing [11–13]. The highly conserved process of

**Funding:** The project described in this publication was supported by the University of Rochester CTSA award number UL1 TR002001 from the National Center for Advancing Translational Sciences (MZ), and award number R01 AI 134058 from the National Institute for Immunology, Allergy, and Infectious Diseases from the National Institutes of Health (MZ, SB). The content is solely the responsibility of the authors and does not necessarily represent the official views of the National Institutes of Health. The funders had no role in study design, data collection and analysis, decision to publish, or preparation of the manuscript.

**Competing interests:** The authors have declared that no competing interests exist.

N-linked glycosylation involves the attachment of a polysaccharide to an asparagine (N) residue facilitated by oligosaccharyltransferase, which recognizes a specific amino acid (AA) sequence known as an N-linked glycosylation sequon [14]. N-linked glycosylation sequons are defined by the AA sequence 'N-!P-S/T' where the first AA is N, the second AA is anything but proline (P), and the third AA is either serine (S) or threonine (T) [15]. Viruses have exploited protein glycosylation mechanisms in host cells to modify their proteins, aiding in viral protein folding, transportation, host receptor binding, and antibody evasion [16]. N-linked glycan trees are anchored at their N residue and can pivot to shield a large protein surface area, sterically blocking antibodies from binding nearby epitopes [17,18]. This effect, known as 'glycan shielding', must be considered when developing monoclonal antibody therapies, vaccination strategies, and antigenic escape predictions.

Influenza hemagglutinin (HA) proteins enable the virus to bind and enter host cells via interaction with sialic acid receptors and are a key target for antibody-mediated immune responses in the host [19]. As influenza virus (IFV) strains circulate, human populations develop immune responses capable of neutralizing closely related IFVs via HA interference, resulting in population immunity that slows the spread of infection [20]. To circumvent this population immunity, IFVs frequently modify their HA proteins [21–24]. These modifications include increases in receptor binding avidity, distal mutations altering HA conformation and epitope accessibility, mutations in epitopes directly altering their structure, and mutations resulting in sequon addition events (SAEs) that allow new glycans to bind and shield vulnerable epitopes [7]. IFV's use of the latter strategy is seldom considered when predicting escape mutations despite the fact that it has occurred dozens of times in the recorded history of human IFVs [25]. Given IFV's propensity for shielding epitopes with glycans, predictive software that identifies future glycosylation sites could improve models for antigenic escape. Here we describe an effective computational method for predicting future glycosylation sites on viral surface proteins. The method identifies which sites on viral proteins are most likely to mutate into N-linked glycosylation sequons based on the region's nucleotide sequence, its surface accessibility, and the historical mutation frequency of residues within the region that would need to change to generate a glycosylation sequon. The method's efficacy is first tested on historic influenza HA proteins and later applied to currently circulating strains.

## Materials and methods

### Program overview

The method for predicting sequon emergence, *Future Sequon Finder* (FSF) [26], was implemented in Java using JDK 21. The general workflow of the program is outlined in Fig 1. It accepts a nucleotide sequence input with the option to include the associated surface accessibility for each AA in the protein acquired from the external surface accessibility prediction software `GetArea` [27] as well as site-specific AA mutation frequencies acquired from the sequence alignment and analysis tool `MEGA` using the Le and Gascuel model [28,29]. The nucleotide input is converted into an AA sequence, cross-referencing the `GetArea` results with the generated AA sequence, and a list of existing N-linked glycosylation sequons is identified, along with lists of sites that are most likely to become glycosylation sequons as a result of genetic drift. Probable future glycosylation sites, or "near-sequons," are sorted into three categories based on the nucleic and amino acid edit distance from a glycosylation sequon and changes in the physical properties of the AA sequence that would emerge from sequon transformation. The categories are as follows:

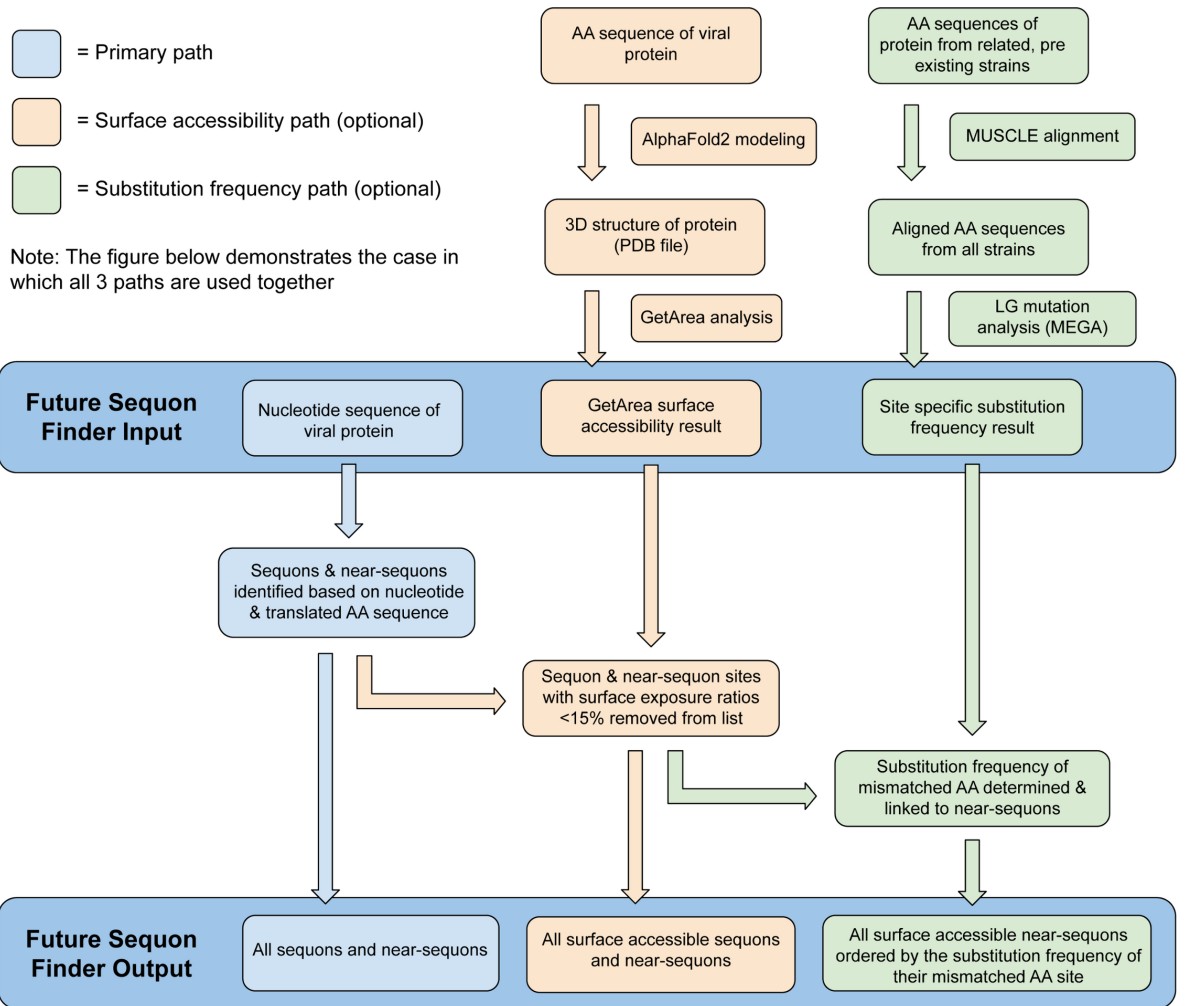

**Fig 1. General workflow for generating outputs with FSF.** The primary analytic pipeline shown in blue is the most basic analytic path in FSF, requiring only an input nucleic acid sequence to generate lists of existing and near sequons. This diagram illustrates a case in which the mutation frequency (green) and surface accessibility (orange) paths are used with the primary path to generate more robust and informative results. When using all pathways, the FSF output will contain a set of binned lists of: all near-sequons and true sequons, all near-sequons and true sequons with exposed surfaces, all near-sequons sorted by mutation frequency (not shown here), and all near-sequons with solvent-exposed surfaces sorted by mutation frequency.

- **Near-sequon** - A trio of amino acids in which one AA must be substituted to generate a glycosylation sequon (AA edit distance = 1)
- **Very near-sequon** - A trio of amino acids in which one nucleotide must be substituted to generate a glycosylation sequon (nucleotide edit distance = 1)
- **Ultra near-sequon** - A very near-sequon wherein the mismatched AA has the same charge, polarity, and hydrophobicity as the correct AA

If surface accessibility data is included with the sequence input, all near-sequons will be sorted by the surface exposure ratio of the first AA in the sequon triad. All near-sequons (including 'very' and 'ultra' near-sequons) with exposure ratios exceeding 15% are tagged as near-sequons with exposed surfaces, meaning they are adequately exposed for glycan attachment. While glycans are covalently bound to proteins before folding, we assume that a sequon

in an unexposed region would either remain unglycosylated, thus providing no selective advantage, or become glycosylated and disrupt proper folding of the protein. If the user has chosen to incorporate site-specific mutation frequency data, near-sequons in each category with exposed surfaces will be sorted again based on the relative mutation frequency of the site where their mismatched AA is located. We assumed as a premise that if a near-sequon was only one AA away from becoming a true sequon, and that AA position was subject to frequent observable mutations in the history of related IFVs, that site was more likely to mutate again and generate a functional sequon. After each category is sorted, the rankings of all near-sequons are displayed to the user with those whose mismatched AA is most likely to mutate ranked at the top of the list.

## Sidechain exposure ratio analysis

To implement the surface accessibility path in FSF, the nucleotide sequences of interest were first translated into AAs. 3D protein models were then generated from the AA sequences and stored as PDB files. ColabFold v1.5.5 (AlphaFold2 using MMseqs2) was used in this analysis to generate all 3D protein structures [30,31]. After the PDB files were generated, the sidechain exposure ratios were determined using `GetArea`, chosen due to its accessibility and ease of use [27]. A 1.4Å probe radius was used for all predictions, the approximated radius of a water molecule.

FSF was designed to receive `GetArea` outputs, assigning a sidechain exposure ratio to each AA found in both the `GetArea` result and the translated nucleotide sequence input by the user. All AA/position identifier pairings from the `GetArea` result are compared to those same pairings generated from the converted nucleotide sequence input. If any mismatches between the `GetArea` sequence and the user input sequence are detected, an error is thrown indicating that the sidechain exposure ratios could not be aligned. Once sidechain exposure ratios have been aligned to the converted AA sequence, FSF iterates through all categories of near-sequons previously determined from the nucleic acid sequence. The program categorizes near-sequons based on whether the sidechain exposure ratio of the first AA in the triad (N, or the residue destined to become N) exceeds 15%. A cutoff of 15% was chosen given the observation that existing sequons seldom support glycans when the asparagine sidechain exposure ratios are less than 20% in HA proteins as found for multiple influenza strain HAs by Altman et al. [25]. The additional 5% leniency was included to account for minor errors in 3D protein structure generation and conformational changes that may occur during the transition from near-sequon to sequon.

## Site-specific mutation rate analysis

To implement the site-specific mutation frequency path in FSF, a list of nucleotide sequences relating to and predating the sequence of interest must first be obtained. These sequences provide evolutionary context for the sequence of interest, showing which regions are most likely to change over time. We excluded sequences with the following characteristics: uncertain nucleotides (any character other than A, T, G, or C), any AA insertions (HA sequences encoding more than 566 AAs), any repeat sequences sharing the same name, and any sequences from strains succeeding our strain of interest. Sequences were curated to ensure they began with the correct start codon, eliminating reading frame ambiguity for translation. Sequences that did not begin with a start codon were eliminated. The remaining nucleotide sequences were converted into AAs, exported in a FASTA file, and then aligned using the multiple sequence alignment tool, `MUSCLE` [32]. The aligned sequences were saved as a .meg file and then imported into the Molecular Evolutionary Genetics Analysis (`MEGA`) tool [28].

Site-specific mutation rates were calculated in `MEGA` using the Le and Gascuel model with 8 gamma distribution categories [29].

The `MEGA` output was then imported into FSF, which assigned relevant site-specific mutation values to each predetermined near-sequon based on the AA position within that near-sequon that would need to mutate to generate a functional N-linked glycosylation sequon. For example, if near-sequon "NTK" was identified with AAs in positions 1, 2, and 3 respectively, FSF would recognize that the "K" AA in position 3 must change to generate a sequon. FSF will then associate the site-specific mutation frequency for AA site 3 with that near-sequon. Near-sequons are then ranked within their classification categories based on their associated mutation frequencies from most mutagenic to least.

## Results

### Predicting historic IFV sequon addition events

To test FSF's efficacy, we used a set of historic influenza strains where SAEs were observed in closely related subsequent strains, as described by Altman et al. [25]. For this analysis, 20 strains of H1N1 IFV and 18 strains of H3N2 IFV relevant to SAEs were selected. Most sequences were selected from S3 Fig of Altman et al., who previously determined the phylogenetic relationships among prominent historic IFV strains associated with SAEs [25]. Exceptions to this include A/NewYorkCity/2/1918, A/Tonga/14/1984, A/Memphis/1/1984, A/Colorado/14/2015, A/Virginia/22/2016, and A/Netherlands/10100/2024 for H1N1, and A/Wyoming/11/2014, A/Germany/13247/2022, A/Bangkok/P3993/2023, A/Netherlands/ 10098/2024 for H3N2 which were added to fill temporal and phylogenetic gaps in the dataset. The added strains were selected from the Bacterial and Viral Bioinformatics Resource Center (BV-BRC) [33] or GISAID's EpiFlu database [34]. To ensure these strains were representative of common glycosylation sequon and near-sequon patterns, they were compared to other circulating strains from the same time period, avoiding rare variants. The strains, sequences, and their sources can be found in S1 Appendix.

Initially, FSF was designed to only consider the genetic composition of HA sequences. Following the primary program path in Fig 1, FSF was able to analyze the selected 20 H1N1 and 18 H3N2 strains with their nucleotide sequences. The nucleic acid sequences for the HA proteins of all 38 selected strains were obtained from the BV-BRC [33] and GISAID's Epi-Flu database [34]. Sites that would become glycosylation sequons at some point in the IFV timeline were tracked over time. Their status as existing sequons, near-sequons, very near-sequons, and ultra near-sequons was recorded (Table 1). We focused on sequon alterations in the H1 head domain, where most N-linked glycosylation changes occur. Sequons and near-sequons outside of this region were omitted to decrease the total number of near-sequons in the FSF outputs.

Considering the potential genetic disparity between IFV strains from consecutive years, a phylogenetic analysis of the same IFV HA sequences was conducted. This strategy avoids pairing distantly related sequences and captures the impact of genetic drift, the primary means by which IFVs adapt to escape herd immunity [7,35], on sequon emergence. Phylogenetic trees containing all selected strains for H1N1 and H3N2 were constructed and are displayed in Fig 3. These trees were used to determine the branch on which each SAE occurred. Assuming parsimony, there were 19 SAEs in the H1N1 lineage and 13 SAEs in the H3N2 lineage that ultimately rose to prominence, persisting for ≥2 years and successfully spreading to multiple countries. It is worth noting that other SAEs have occurred in recorded human H1N1 and H3N2 HA sequences; however, these glycosylation sequons appeared in single countries or disappeared within 1–2 years. This analysis only considers SAEs that rose

**Table 1. Temporally organized FSF results in the context of historically glycosylated HA regions.** Historic H1N1 and H3N2 IFV strains have been temporally organized to illustrate trends in glycosylation patterns over time, as well as trends in near-sequon occurrence at these critical sites. The letters in each AA site column are the single-letter identifiers for the AA found at the given position in the given strain. These letters also represent the first AA of the sequon (or near-sequon) triad (the "N" position of N-!P-S/T). Characterized antigenic sites are indicated in the top row of the table. Near-sequons identified by FSF are indicated by varying shades of orange, with darker shades representing sites that more closely resemble glycosylation sequons. The emboldened letters in the darkest shade of orange represent true N-linked glycosylation sequons. (Top) Temporally organized H1N1 IFVs. (Bottom) Temporally organized H3N2 IFVs.

| Amino Acid Positions of Glycosylation Sites | | | | | | | | | | | | Near-Sequon |
| | Cb | | Sa | | | | | | | | | Very near-sequon |
| H1N1 Strain | 71 | 90 | 104 | 142 | 144 | 172 | 177 | 179 | 286 | 293 | Year | Ultra near-Sequon / **Sequon** |
| --- | --- | --- | --- | --- | --- | --- | --- | --- | --- | --- | --- | --- |
| A/NewYorkCity/2/1918 | K | A | N | N | E | G | K | S | D | N | 1918 | |
| A/Wilson-Smith/1933 | K | A | N | N | N | G | K | N | N | N | 1933 | |
| A/PuertoRico/8/1934 | K | V | N | N | N | E | K | K | N | N | 1934 | |
| A/Melbourne/JY2/1935 | K | A | N | K | N | E | K | S | N | N | 1935 | |
| A/Hickox/JY2/1940 | K | K | N | N | N | D | N | N | N | D | 1940 | |
| A/Bellamy/JY2/1942 | K | E | N | K | N | D | N | N | N | N | 1942 | |
| A/Weiss/JY2/1943 | K | E | N | K | N | D | N | K | N | D | 1943 | |
| A/Cameron/JY2/1946 | K | K | N | K | N | D | N | N | N | D | 1946 | |
| A/FortMonmouth/1-JY2/1947 | K | K | N | K | N | D | K | S | N | D | 1947 | |
| A/Malaysia/JY2/1954 | K | N | N | K | N | N | S | S | N | D | 1954 | |
| A/Denver/JY2/1957 | N | N | N | N | Del | N | N | S | N | D | 1957 | |
| A/USSR/90/1977 | K | K | N | K | N | N | N | S | N | D | 1977 | |
| A/Tonga/14/1984 | K | K | N | N | N | K | N | S | N | D | 1984 | |
| A/Memphis/1/1987 | N | K | N | N | T | N | N | S | N | D | 1986 | |
| A/Texas/36-JY2/1991 | N | K | N | N | T | N | N | S | N | D | 2006 | |
| A/Brisbane/59/2007 | N | K | N | N | T | N | N | S | N | D | 2007 | |
| A/California/07/2009 | K | A | N | N | D | G | K | S | D | N | 2009 | |
| A/Colorado/14/2015 | K | A | N | N | D | G | K | S | D | N | 2015 | |
| A/Virginia/22/2016 | K | A | N | N | D | G | K | N | D | N | 2016 | |
| A/Netherlands/10070/2024 | Q | A | N | N | D | G | K | N | D | N | 2024 | |

| Amino Acid Positions of Glycosylation Sites | | | | | | | | | | | | Near-Sequon |
| | C | E | | | A | | | | | C | | | Very near-sequon |
| H3N2 Strain | 61 | 79 | 97 | 110 | 138 | 142 | 149 | 160 | 174 | 181 | 262 | 292 | Year / Ultra near-Sequon / **Sequon** |
| --- | --- | --- | --- | --- | --- | --- | --- | --- | --- | --- | --- | --- | --- |
| A/Hong Kong/1/1968 | S | D | N | F | T | T | N | G | G | N | N | T | 1968 |
| A/Udorn/307/1972 | S | D | N | F | S | T | N | D | G | N | N | T | 1972 |
| A/Victoria/3/1975 | S | N | N | F | N | N | N | D | G | N | N | T | 1975 |
| A/Bangkok/1/1979 | S | N | N | F | N | N | S | D | E | N | N | T | 1979 |
| A/Philippines/2/1982 | S | N | N | F | N | N | S | N | E | N | N | T | 1982 |
| A/Beijing/32/1992 | S | N | N | Y | N | N | D | V | E | N | N | T | 1992 |
| A/Harbin/15/1992 | S | N | N | Y | N | N | D | V | E | N | N | N | 1992 |
| A/Nanchang/933/1995 | S | N | N | Y | N | N | D | V | E | N | K | N | 1995 |
| A/Sydney/5/1997 | S | N | N | Y | N | N | N | I | K | N | N | K | 1997 |
| A/Moscow/10/1999 | S | N | N | Y | N | N | N | N | E | N | N | K | 1999 |
| A/Ulan Ude/01/2000 | S | N | N | Y | N | N | N | I | N | N | N | K | 2000 |
| A/Brisbane/10/2007 | S | N | N | Y | N | N | N | N | K | N | N | K | 2007 |
| A/Wisconsin/15/2009 | S | N | N | Y | N | N | N | K | N | N | N | K | 2009 |
| A/Victoria/361/2011 | N | N | N | Y | N | N | N | N | N | N | N | K | 2011 |
| A/Wyoming/11/2014 | N | N | N | Y | N | N | N | S | N | N | N | K | 2014 |
| A/Germany/13247/2022 | N | N | N | N | N | N | N | S | N | N | N | R | 2022 |
| A/Bangkok/P3993/2023 | N | N | N | N | N | N | N | S | N | N | N | K | 2023 |
| A/Netherlands/10098/2024 | N | N | N | N | D | N | N | S | N | N | N | E | 2024 |

to prominence as previously defined. Preexisting strains closely related to the SAE strain were selected and analyzed with FSF. Table 2 displays the results of this phylogenetic-based analysis. To better visualize the disparity between HA sequences within IFV groups over time, AA edit distance heatmaps were generated using randomly selected HA strains from the H1N1 and H3N2 master lists (Fig 2).

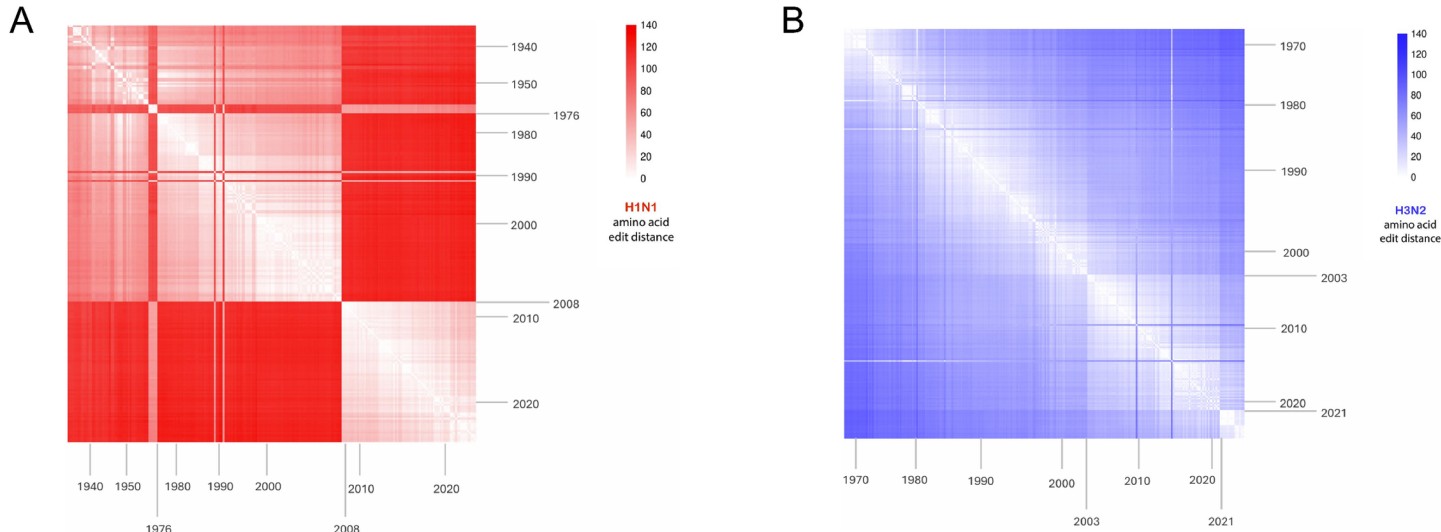

**Fig 2. Mapping edit distance between IFV HAs over time.** To better understand the roles of antigenic shift and antigenic drift in H1N1 and H3N2 HA evolution, edit distances between HA AA sequences were calculated and visualized in RStudio. All HA sequences were randomly selected from the master lists comprised of 32,000 H1N1 sequences and 40,000 H3N2 sequences, with a maximum of 5 sequences drawn per year to prevent overrepresentation of modern strains. (**A**) H1N1 HA protein AA edit distances from 1918–2024. (**B**) H3N2 HA protein AA edit distances from 1968–2024.

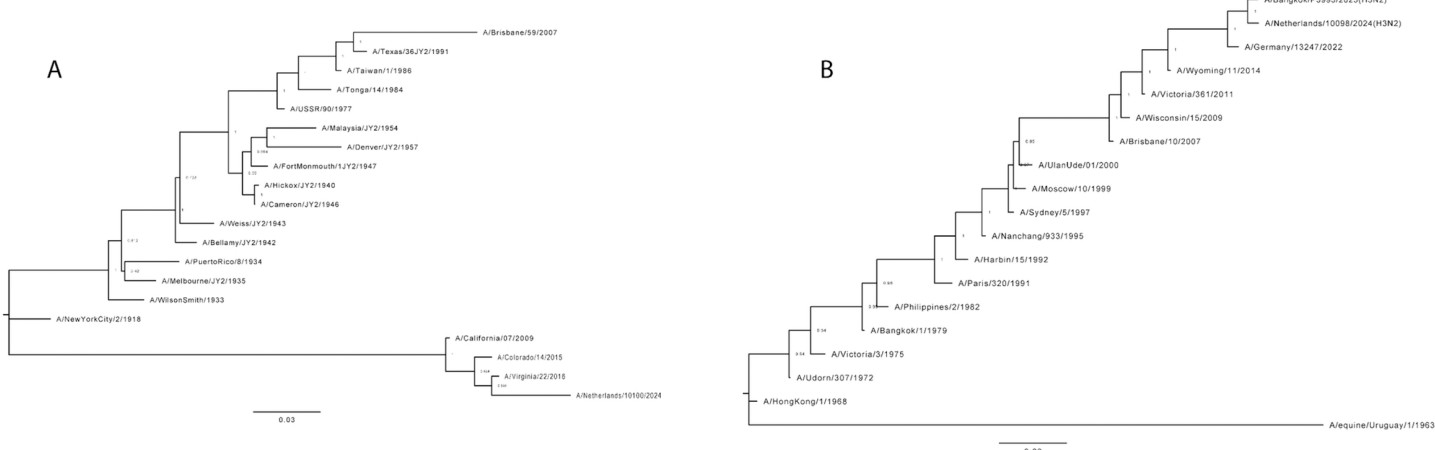

**Fig 3. Phylogenetic trees containing selected H1N1 and H3N2 IFV HAs.** HA nucleotide sequences for the 20 H1N1 and 17 H3N2 IFVs chosen for historical analysis were aligned and used to generate maximum likelihood phylogenetic trees in MEGA [28]. Bootstrap values (500 replicates) are displayed at each branch node as proportions. (**A**) Maximum likelihood tree of selected H1N1 IFVs. (**B**) Maximum likelihood tree of selected H3N2 IFVs. A/equine/Uruguay/1/1963 was used as an outgroup to root the trees.

Through the phylogenetic analysis of historic IFV strains with FSF, we found that N-linked glycosylation sites on HA proteins almost always arise due to single nucleotide mutations. Of the observed 30 SAEs in the historic IFV selection, 28 arose due to single nucleotide mutations. Of the two that did not, one appeared to be the result of a deletion that occurred in A/Denver/JY2/1957 (compared to A/Malaysia/JY2/1954) at AA position 144, followed by two single nucleotide changes, one in AA 142 and the other in AA 145, which occupied position 144 following the deletion event. The other appeared to be due to two single nucleotide

**Table 2. FSF site designations of critical sites from closely related IFV strains circulating before SAEs. All AA sites listed in this table became N-linked glycosylation sequons in the strains specified in the "SAE Strain" column. Phylogenetically related IFV strains circulating before the SAE have been selected and the FSF designation of the site that would become glycosylated is presented in the rightmost column. Related strains were selected if they predated the strain in which the glycosylation sequon arose and were the closest preexisting relative on the generated phylogenetic tree shown in Fig 3.**

| Group | Strain name | Site of SAE | Strain circulating prior to SAE | Near-sequon Designation |
|---|---|---|---|---|
| H1N1 | A/WilsonSmith/1993 | 179 | A/NewYorkCity/2/1918 | Ultra near-sequon |
| | A/WilsonSmith/1993 | 286 | A/NewYorkCity/2/1918 | Not detected |
| | A/Melbourne/JY2/1935 | 144 | A/PuertoRico/8/1934 | Ultra near-sequon |
| | A/Bellamy/JY2/1942 | 104 | A/Melbourne/JY2/1935 | Very near-sequon |
| | A/Bellamy/JY2/1942 | 179 | A/Melbourne/JY2/1935 | Ultra near-sequon |
| | A/Hickox/JY2/1940 | 179 | A/Melbourne/JY2/1935 | Ultra near-sequon |
| | A/FortMonmouth/1-JY2/1947 | 144 | A/Cameron/JY2/1946 | Ultra near-sequon |
| | A/Malaysia/JY2/1954 | 90 | A/FortMonmouth/1-JY2/1947 | Very near-sequon |
| | A/Denver/JY2/1957 | 104 | A/Malaysia/JY2/1954 | Very near-sequon |
| | A/Denver/JY2/1957 | 142 | A/Malaysia/JY2/1954 | Not detected |
| | A/Denver/JY2/1957 | 172 | A/Malaysia/JY2/1954 | Very near-sequon |
| | A/Denver/JY2/1957 | 177 | A/Malaysia/JY2/1954 | Ultra near-sequon |
| | A/USSR/90/1977 | 144 | A/Cameron/JY2/1946 | Ultra near-sequon |
| | A/USSR/90/1977 | 172 | A/Cameron/JY2/1946 | Very near-sequon |
| | A/USSR/90/1977 | 177 | A/Cameron/JY2/1946 | Ultra near-sequon |
| | A/Memphis/1/1987 | 71 | A/Tonga/14/1984 | Very near-sequon |
| | A/Memphis/1/1987 | 142 | A/Tonga/14/1984 | Ultra near-sequon |
| | A/California/07/2009 | 293 | A/NewYorkCity/2/1918 | Very near-sequon |
| | A/Virginia/22/2016 | 179 | A/Colorado/14/2015 | Ultra near-sequon |
| Group | Strain Name | Site of SAE | Strain circulating prior to SAE | Near-sequon Designation |
| H3N2 | A/Victoria/3/1975 | 79 | A/Udorn/307/1972 | Very near-sequon |
| | A/Victoria/3/1975 | 142 | A/Udorn/307/1972 | Ultra near-sequon |
| | A/Philippines/2/1982 | 160 | A/Bangkok/1/1972 | Very near-sequon |
| | A/Philippines/2/1982 | 262 | A/Bangkok/1/1972 | Ultra near-sequon |
| | A/Harbin/15/1992 | 292 | A/Beijing/32/1992 | Ultra near-sequon |
| | A/Sydney/5/1997 | 138 | A/Nanchang/933/1995 | Very near-sequon |
| | A/Sydney/5/1997 | 149 | A/Nanchang/933/1995 | Very near-sequon |
| | A/Sydney/5/1997 | 262 | A/Nanchang/933/1995 | Very near-sequon |
| | A/Moscow/10/1999 | 160 | A/Sydney/5/1997 | Very near-sequon |
| | A/Victoria/361/2011 | 61 | A/Wisconsin/15/2009 | Ultra near-sequon |
| | A/Victoria/361/2011 | 160 | A/Wisconsin/15/2009 | Very near-sequon |

mutations in A/WilsonSmith/1933 (compared to A/NewYorkCity/2/1918), one in AA codon 286 and the other in AA 288. In both cases, these mutations may have arisen simultaneously to generate glycosylation sequons. Alternatively, it could be that more closely related strains containing near-sequons at these sites were never sequenced given the limited number of HA sequences available from the early 20th century. The notion that N-linked glycosylation sequons almost always arise due to single nucleotide mutations coincides with the finding that antigenic diversity in IFV HAs is primarily driven by the gradual accumulation of single base pair mutations [36,37]. Though unsurprising, this finding allows for more accurate sequon predictions given that sites requiring more than a single base mutation can be eliminated from consideration. While it is clear that the ultra and very near-sequon categories have a significantly different proportion of SAEs than the near-sequon category, $\chi^2$ (2, N = 711) = 16.736, p = 0.0002, the difference between the ultra and very near-sequon categories is less apparent, $\chi^2$ (1, N = 274) = 2.6353, p = 0.1045. More data is needed to determine if this difference is statistically significant.

## Side chain surface exposure and mutation frequency analysis improve prediction

While FSF had proven to be effective at tagging sites as near-sequons in strains circulating before SAEs, it would also tag many sites that would never become sequons. To eliminate some of these false positives, the AA-specific side chain surface exposure analysis method was implemented. The sidechain exposure ratios for each IFV strain were determined using AlphaFold2 and `GetArea`. HAs were analyzed in their monomeric forms using `GetArea` to determine sidechain exposure.As a result, the number of exposed sites is likely overestimated compared to trimerized HAs, given that fewer regions of the HA are exposed in its oligomeric state. Although glycosylation precedes oligomerization, glycans that sterically hinder HA trimerization would likely be selected against. Measuring the surface accessibility of AAs in conformationally closed, trimerized HAs would likely reduce the number of exposed sequons and improve prediction accuracy, though this analysis has not yet been performed.

The site-specific mutation frequency analysis method was implemented to further organize the remaining near-sequons with exposed sidechains. To determine mutation frequencies, HA nucleotide sequences were obtained from BV-BRC and GISAID (32,000 H1N1 sequences, 40,000 H3N2 sequences) to form H1N1 and H3N2 master lists. The mutation frequencies were calculated separately for the analysis of every strain preceding an SAE using only strains that predated or coincided with the analysis strain. For example, site-specific mutation frequencies for A/Udron/307/1972 (H3N2) were calculated only using strains from the H3N2 master set that circulated before 1972. Due to computational restrictions, if a single year had more than 500 strains, 500 were randomly selected to be used in the analysis. For years with strain numbers below the cutoff, all strains were used. The number of IFV HA genomes obtained for each year is shown in Fig 4. At this stage, FSF was using all three pathways outlined in Fig 1 to generate the historic results shown in Table 3. Strain comparisons matched those used in the phylogenetic analysis.

## Predicting sequon emergence in swine H1N1 strains

We next examined the predictive capacity of FSF by testing its ability to predict the emergence of glycosylation sequons in swine H1N1 strains. Six thousand swine H1N1 sequences collected from 1931–2022 were obtained from the BV-BRC [33] and pre-processed using a custom Java program designed to identify SAE strains. Once identified, a maximum-likelihood phylogenetic tree containing 15 SAE strains and 700 other H1N1 swine strains (a maximum of 25 strains selected from each year, 1931–2022) was generated, and close relatives to the SAE strains were identified. Those relatives predating the SAE were analyzed with FSF with their accompanying sidechain exposure and mutation frequency data as described for the human influenza strains. Unlike our analysis of human IFVs above, sequons did not need to meet the definition of prominence for their SAE strains to be included in this analysis. The only swine SAE strains to be excluded from the analysis were those with no close ancestors, defined as strains on isolated branches of the phylogenetic tree. The results of this analysis are displayed in Table 4.

In the case of the swine influenza dataset, near-sequon designations, along with sidechain exposure ratios, were effective predictive tools. Fourteen of 15 SAEs were tagged as near-sequons with exposed surfaces, with 12 of those 14 being very near-sequons. Site-specific mutation frequency proved less effective in this dataset, with 50% of very near-sequons being ranked in the bottom half of the very near-sequon pool, compared to 15% in the human H1N1 dataset and 23% in the human H3N2 dataset. There are many differences between IFV circulation in swine and human populations that may account for this disparity. Swine herds

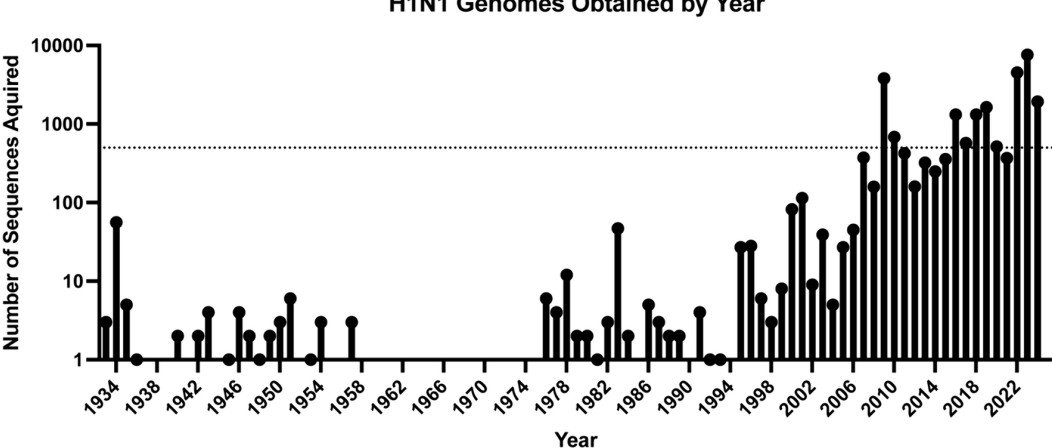

**Fig 4. Influenza genomes obtained by year.** These histograms show the total number of sequences in the H1N1 and H3N2 master lists acquired from the BV-BRC [33] and GISAID's EpiFlu database [34] for each year. The dotted line at Y = 500 represents the aforementioned cutoff applied when calculating site-specific mutation frequencies.

are often tightly regulated and have limited exposure to other herds, in contrast to humans who frequently encounter other individuals worldwide [38]. This may explain why SAEs in the swine dataset rarely rise to prominence compared to those in the human dataset. The isolation of swine herds may also drive local adaptation of swine IFV, where mutation patterns are shaped by specific environmental pressures unique to each herd. As a result, mutation frequencies observed in one swine population may not align with those in others, which could explain the reduced effectiveness of the mutation frequency analysis in this dataset.

## Predicting future IFV sequon addition events

Following the predictive success FSF displayed with the historic IFV datasets, it was time to generate future predictions from currently circulating strains. Ten currently circulating IFV strains were selected for both H1N1 and H3N2 from GISAID's EpiFlu database. These strains

**Table 3. Historic FSF predictions with side-chain exposure and site-specific mutation frequency data.** The same closely related H1N1 and H3N2 IFV strains analyzed in Fig 2 were reanalyzed with side-chain exposure and site-specific mutation frequency data. The surface exposure of the first AA in each near-sequon is expressed as a percentage from the GetArea results. The relative mutation rate of the mismatched AA position is provided, where a value of 1 represents the average mutation rate for AAs across the HA protein. The site-specific mutation gamma category placement (out of 8) is displayed, with AAs assigned to category 8 being the most likely to mutate. The certainty with which MEGA assigned sites to their respective gamma categories is also shown as a proportion. Near-sequons within the same HA are based on relative AA mutation rates. Near-sequons with surface exposure ratios ≤15% are excluded from the rankings.

| Group | Related strain circulating prior to SAE | Mismatched AA position | Mismatched AA(s) in Near-Sequon | Mismatched Nucleotide(s) | Sidechain exposure ratio of first AA (%) | Relative substitution rate (mismatched AA site) | Gamma Category | Gamma Category Certainty | Near-sequon designation | Sorted near-sequon ranking | Sorted very near-sequon ranking | Sorted ultra near-sequon ranking | Related strain(s) in which predicted SAE occurred |
|---|---|---|---|---|---|---|---|---|---|---|---|---|---|
| H1N1 | A/NewYorkCity/2/1918 | 179 | (S)KS | A(G)C AAG TCC | 61.90% | NA | NA | NA | Ultra near-sequon | NA | NA | NA | A/WilsonSmith/1933 |
| | A/NewYorkCity/2/1918 | 286, 288 | (D)A(P) | (G)AC GCA (C)CA | 62.80% | NA | NA | NA | Not detected | NA | NA | NA | A/WilsonSmith/1933 |
| | A/NewYorkCity/2/1918 | 293 | NT(K) | AAC ACG A(A)G | 66.70% | NA | NA | NA | Very near-sequon | NA | NA | NA | A/California/07/2009 |
| | A/PuertoRico/8/1934 | 146 | NT(N) | AAC ACA A(A)C | 41.50% | 5.0823 | 8 | 0.823914226 | Ultra near-sequon | 4 of 34 | 3 of 18 | 1 of 5 | A/Melbourne/IY2/1935 |
| | A/Melbourne/IY2/1935 | 106 | NG(I) | AAT GGA A(T)A | 24.30% | 2.9135 | 8 | 0.394955256 | Very near-sequon | 8 of 30 | 7 of 17 | NA | A/Bellamy/IY2/1942 |
| | A/Melbourne/IY2/1935 | 179 | (S)NS | A(G)C AAT TCC | 63.30% | 5.6767 | 8 | 0.995251217 | Ultra near-sequon | 2 of 30 | 1 of 17 | 1 of 4 | A/Bellamy/IY2/1942, A/Hickox/IY2/1940 |
| | A/Cameron/IY2/1946 | 146 | NI(N) | AAC ATA A(A)C | 17.00% | 5.5807 | 8 | 0.999900019 | Ultra near-sequon | 1 of 25 | 1 of 11 | 1 of 3 | A/FortMonmouth/1-IY2/1947, A/USSR/90/1977 |
| | A/Cameron/IY2/1946 | 172 | (D)GS | (G)AT GGC TCA | 80.30% | 5.5802 | 8 | 0.999756667 | Very near-sequon | 2 of 25 | 2 of 11 | NA | A/USSR/90/1977 |
| | A/Cameron/IY2/1946 | 179 | NL(N) | A(A)T CTG AAC | 65.60% | 5.5642 | 8 | 0.995627834 | Ultra near-sequon | 3 of 25 | 3 of 11 | 2 of 3 | A/USSR/90/1977 |
| | A/FortMonmouth/1-IY2/1947 | 90 | (K)RS | AA(G) AGA TCA | 62.40% | 4.9818 | 8 | 0.829863191 | Very near-sequon | 6 of 27 | 5 of 13 | NA | A/Malaysia/IY2/1954 |
| | A/Malaysia/IY2/1954 | 106 | NG(I) | AAT GGA A(T)A | 23.30% | 5.9651 | 8 | 0.990820069 | Very near-sequon | 3 of 25 | 3 of 13 | NA | A/Denver/IY2/1957 |
| | A/Malaysia/IY2/1954 | 142, 144 | (K)H(N) | AA(A) CAC AAC A(T)A | 79.80% | NA | NA | NA | Not detected | NA | NA | NA | A/Denver/IY2/1957 |
| | A/Malaysia/IY2/1954 | 174 | NG(L) | AAT GGC T(T)A | 95.20% | 1.1588 | 7 | 0.338051447 | Very near-sequon | 12 of 25 | 7 of 13 | NA | A/Denver/IY2/1957 |
| | A/Malaysia/IY2/1954 | 177 | (S)LS | A(G)T CTG AGC | 71.20% | 5.9934 | 8 | 0.997545073 | Ultra near-sequon | 2 of 25 | 2 of 13 | 2 of 3 | A/Denver/IY2/1957 |
| | A/Tonga/14/1984 | 71 | (K)CS | AA(A) TGC AGC | 79.40% | 1.1287 | 6 | 0.317652827 | Very near-sequon | 16 of 24 | 7 of 11 | NA | A/Memphis/1/1987 |
| | A/Tonga/14/1984 | 144 | NH(N) | AAC CAC A(A)C | 69.00% | 5.8352 | 8 | 0.996265704 | Ultra near-sequon | 4 of 24 | 3 of 11 | 2 of 2 | A/Memphis/1/1987 |
| | A/Colorado/14/2015 | 179 | (S)QS | A(G)C CAA TCC | 59.20% | 4.9728 | 8 | 0.98307399 | Ultra near-sequon | 6 of 28 | 1 of 8 | 1 of 1 | A/Virginia/22/2016 |
| Group | Related strain circulating prior to SAE | Mismatched AA position | Mismatched AA(s) in Near-Sequon | Mismatched Nucleotide(s) | Sidechain exposure ratio of first AA (%) | Relative substitution rate (mismatched AA site) | Gamma Category | Gamma Category Certainty | Near-sequon designation | Sorted near-sequon ranking | Sorted very near-sequon ranking | Sorted ultra near-sequon ranking | Related strain(s) in which predicted SAE occurred |
| H3N2 | A/Udorn/307/1972 | 79 | (D)CT | (G)AC TGC ACA | 31.40% | 3.6715 | 8 | 0.781462859 | Very near-sequon | 1 of 34 | 1 of 15 | NA | A/Victoria/3/1975 |
| | A/Udorn/307/1972 | 142 | (T)WT | A(C)T TGG ACT | 95.40% | 1.7177 | 7 | 0.268087907 | Ultra near-sequon | 16 of 34 | 5 of 15 | 3 of 5 | A/Victoria/3/1975 |
| | A/Bangkok/1/1972 | 160 | (D)NS | (G)AT AAC AGT | 82.80% | 3.8881 | 8 | 0.740567561 | Very near-sequon | 1 of 32 | 1 of 17 | NA | A/Philippines/2/1982 |
| | A/Bangkok/1/1972 | 264 | NS(N) | AAT AGT A(A)T | 46.60% | 1.2865 | 6 | 0.256193369 | Ultra near-sequon | 19 of 32 | 11 of 17 | 4 of 6 | A/Philippines/2/1982 |
| | A/Beijing/32/1992 | 292 | (T)CS | A(C)C TGC AGT | 100.00% | 3.4277 | 8 | 0.591617143 | Ultra near-sequon | 3 of 30 | 1 of 15 | 1 of 3 | A/Harbin/15/1992 |
| | A/Nanchang/933/1995 | 140 | NE(G) | AAT GAA (G)GC | 63.80% | 5.9513 | 8 | 0.997421383 | Very near-sequon | 2 of 27 | 2 of 12 | NA | A/Sydney/5/1997 |
| | A/Nanchang/933/1995 | 149 | (D)GT | (G)AT GGA ACA | 77.30% | 3.6479 | 7 | 0.541101912 | Very near-sequon | 8 of 27 | 5 of 12 | NA | A/Sydney/5/1997 |
| | A/Nanchang/933/1995 | 262 | (K)ST | A(A)A AGC ACA | 55.60% | 5.9535 | 8 | 0.99802238 | Very near-sequon | 1 of 27 | 1 of 12 | NA | A/Sydney/5/1997 |
| | A/Sydney/5/1997 | 160 | (I)KS | A(T)T AAA AGT | 86.20% | 5.2068 | 8 | 0.90121854 | Very near-sequon | 4 of 25 | 2 of 10 | NA | A/Moscow/10/1999 |
| | A/Wisconsin/15/2009 | 61 | (S)SS | A(G)T TCC TCA | 71.90% | 5.4865 | 8 | 0.999745927 | Ultra near-sequon | 2 of 27 | 1 of 12 | 1 of 1 | A/Victoria/361/2011 |
| | A/Wisconsin/15/2009 | 160 | (K)NS | AA(A) AAC AGT | 91.90% | 5.486 | 8 | 0.999585233 | Very near-sequon | 3 of 27 | 2 of 12 | NA | A/Victoria/361/2011 |
| | A/Victoria/361/2011 | 176 | NF(K) | AAC TTC A(A)A | 82.60% | 0.8454 | 5 | 0.464610044 | Very near-sequon | 12 of 21 | 5 of 8 | NA | A/Wyoming/11/2014 |
| | A/Germany/13247/2022 | 112 | NS(N) | AAC AGC A(A)C | 64.40% | 1.4552 | 6 | 0.784152388 | Ultra near-sequon | 13 of 22 | 7 of 9 | 2 of 2 | A/Bangkok/P3993/2023 |

**Table 4. Historic Swine H1N1 FSF predictions.** This table presents FSF results for swine IFVs closely related to SAE strains, incorporating side-chain exposure and site-specific mutation frequency data. The layout of this table is identical to Table 3. Consistent with the human IFV analysis, near-sequons with surface exposure ratios <15% are excluded from the rankings.

| Group | Related strain circulating prior to SAE | Mismatched AA position | Mismatched AA(s) in Near-Sequon | Mismatched Nucleotide(s) | Sidechain exposure ratio of first AA (%) | Relative substitution rate (mismatched AA site) | Gamma Category | Gamma Category | Near-sequon designation | Sorted near-sequon ranking | Sorted very near-sequon ranking | Sorted ultra near-sequon ranking | Related strain(s) in which predicted SAE occurred |
|---|---|---|---|---|---|---|---|---|---|---|---|---|---|
| | A/swine/Wisconsin/1/1971 | 179 | (S)KS | A(G)C AAA TCC | 63.50% | 0.7173 | 2 | 0.14482759 | Ultra near-sequon | 33 of 34 | 11 of 11 | 2 of 2 | A/swine/Minnesota/24/1975 |
| | A/swine/Minnesota/24/1975 | 295 | NT(K) | AAC ACG A(A)G | 61.60% | 0.8339 | 1 | 0.140842184 | Very near-sequon | 22 of 33 | 8 of 10 | NA | A/swine/Kentucky/1/1976 |
| | A/swine/HongKong/299/1993 | 136 | (K)AS | AA(G) GCA AGT | 31.50% | 1.0901 | 6 | 0.253344634 | Very near-sequon | 17 of 33 | 5 of 10 | NA | A/swine/HongKong/NS143/2000 |
| | A/swine/HongKong/8512/2001 | 73 | NC(N) | AAC TGC A(A)C | 75.10% | 0.6388 | 5 | 0.259383068 | Ultra near-sequon | 30 of 40 | 12 of 18 | 3 of 4 | A/swine/HongKong/NS856/2004 |
| | A/swine/Beienrode/3053/2004 | 277 | NK(G) | AAT AAG (G)GC | 49.60% | 3.7095 | 8 | 0.6063151 | Very near-sequon | 9 of 39 | 2 of 16 | NA | A/swine/Bocholt/5533/2006 |
| | A/swine/HongKong/72/2007 | 203 | NY(R) | AAC TAC (C)GT | 53.50% | 5.1865 | 8 | 0.999999937 | Very near-sequon | 1 of 41 | 1 of 16 | NA | A/swine/HongKong/1559/2008 |
| Swine H1N1 | A/swine/Shandong/1012/2008 | 103 | NS(K) | AAT TCA A(A)A | 87.80% | 5.5669 | 8 | 0.999871778 | Very near-sequon | 3 of 40 | 1 of 20 | NA | A/swine/Shandong/275/2009 |
| | A/swine/Minnesota/SG1317/2008 | 175 | NS(Y) | AAT TCA T(A)C | 96.30% | 0.1249 | 1 | 0.32524472 | Very near-sequon | 29 of 32 | 10 of 12 | NA | A/swine/Mississippi/A01202593/2011 |
| | A/swine/HongKong/1559/2008 | 278 | (A)SS | (G)(C)(C) AGT TCT | 100.00% | 2.3219 | 7 | 0.951529425 | Near-sequon | 10 of 31 | NA | NA | A/swine/Hong_Kong/247/2009 |
| | A/swine/Iowa/A01049239/2010 | 202 | (S)A(D) | A(G)T GCT (G)(A)C | 88.50% | NA | NA | NA | Not detected | NA | NA | NA | A/swine/Italy/325451/2011 |
| | A/swine/Jiangsu/zg7/2010 | 213 | NN(H) | AAT AAT C(A)(T) | 43.20% | 2.6538 | 7 | 0.86960537 | Near-sequon | 11 of 39 | NA | NA | A/swine/Jiangsu/zg11/2011 |
| | A/swine/Molbergen/12127/2010 | 256 | (T)IT | A(C)C ATA ACC | 41.90% | 0.6454 | 5 | 0.445352192 | Ultra near-sequon | 30 of 41 | 13 of 19 | 4 of 4 | A/swine/Viersen/13836/2011 |
| | A/swine/Mississippi/A01202593/2011 | 187 | NQ(K) | AAC CAG A(A)G | 71.80% | 5.3217 | 8 | 0.999983929 | Very near-sequon | 2 of 31 | 1 of 12 | NA | A/swine/Oklahoma/A01733315/2016 |
| | A/swine/NorthCarolina/A01476862/2014 | 148 | NR(G) | AAC AGA (G)GT | 79.40% | 0.3633 | 3 | 0.469381124 | Very near-sequon | 25 of 31 | 7 of 9 | NA | A/swine/NorthCarolina/154074/2015 |
| | A/swine/Italy/78383/2020 | 241 | NQ(A) | AAC CAA (G)CA | 85.70% | 2.4509 | 7 | 0.980488226 | Very near-sequon | 7 of 38 | 3 of 18 | NA | A/swine/Italy/9168127/2022 |

were selected semi-randomly, with each strain obtained from a different nation to cover a wider range of circulating influenza geographic diversity. The selected strains include:

- **H1N1** - A/Netherlands/10114/2024, A/Curacao/10079/2024, A/Aragon/102/2024, A/Bayern/33/2024, A/SriLanka/32/2024, A/Bangkok/P479/2024, A/Timis/563634/2024, A/Norway/01013/2024, A/NorthCarolina/14611/2024, A/UnitedKingdom/14762/2024
- **H3N2** - A/Curacao/10082/2024, A/Canberra/14/2024, A/Bangkok/P458/2024, A/ Hungary/1/2024, A/Netherlands/10110/2024, A/Norway/00732/2024, A/Victoria/12/2024, A/Philippines/2/2024, A/Yekaterinburg/2/2024, A/Oklahoma/14622/2024

Each chosen strain was analyzed using all FSF pathways. Sidechain exposure ratios were calculated for each strain individually and mutation frequencies were calculated with the same HA sequence dataset and 'pick 500' approach used for the historic strain analysis. For H1N1, mutation frequencies were calculated using strains from all years (1918-2024). Mutation frequencies were recalculated with strains from 2009-2024 to determine if there was a noteworthy difference between the resultant ranking of near-sequons given the large phylogenetic distance between modern H1N1 strains and those circulating pre-2009. For H3N2 viruses, the mutation frequency was calculated once using strains from all years (1968-2024). All accurate SAE predictions in the historic human dataset were predicted as very near-sequons or ultra near-sequons, so it is logical to prioritize these categories when predicting future SAEs. Consequently, exposed very near-sequons found in the circulating IFV strains were assigned an overall rank based on the relative mutation rate of their mismatched AA site. The frequency of the very near-sequon sites in the analyzed IFVs was reported but not factored into the overall ranking. Final rankings were determined by site-specific mutation frequency alone. The summarized results of these future sequon predictions are displayed in Table 5.

## Discussion

The temporal analysis of historic H3N2 IFVs supports the notion that the HA glycosylation patterns of currently circulating strains are most closely related to the HAs of strains circulating in the previous year (Figs 2 and 3). The temporal analysis of H1N1 IFVs revealed a similar pattern with some exceptions, given that strains closely related to distant IFVs occasionally reemerged and replaced the predominant strains. The most dramatic example was the emergence of A(H1N1)pdm09 type IFVs from swine reservoirs in 2009, whose HA proteins and glycosylation patterns were more closely related to those from the 1976 swine outbreak than any circulating in the prior decade. This demonstrates the importance of monitoring for the re-emergence of IFVs propagating in human or animal reservoirs at low levels. However, our results suggest that for both H1N1 and H3N2, the HA sequences of prominent circulating strains are the best FSF inputs for predicting sequons that will arise in the near future.

In both the temporal and phylogenetic analyses of historic IFVs, falsely predicted sequons greatly outnumbered accurate predictions. Consequently, the reduction and sorting of near-sequon pools became a priority, leading to the implementation of the two optional pathways shown in Fig 1. The sidechain exposure ratio analysis of individual AA sites greatly narrowed the near-sequon pools. A typical implementation of the surface accessibility pathway reduces HA near-sequon pools by 35–45% (∼50 near-sequons reduced to ∼30). In the analysis of historic human and swine IFVs, no near-sequon that would subsequently become a sequon was excluded by the 15% sidechain exposure cutoff, though some were close (within 2%). To be safe in future analysis, it may be beneficial to slightly lower this cutoff.

**Table 5. Future H1N1 and H3N2 N-linked glycosylation sequon locations predicted by FSF.** Each table summarizes the top site predictions from 10 individual FSF runs using currently circulating IFV HAs. Overall near-sequon rankings were determined by site-specific mutation rates, with higher rates corresponding to higher overall rankings. Near-sequon frequencies among the analyzed strains are displayed as proportions, alongside the average side-chain exposure ratios of the first amino acids (AAs) in the near-sequon triads. The mutation rates for the mismatched AA positions and their gamma category assignments are also presented, along with the near-sequon distinction of the site and its ranking among other near-sequons based on mutation rates. (Top) Future H1N1 sequon locations predicted by FSF using a mutation frequency calculation based on a random selection of ≤ 500 H1N1 HA sequences per year for all years from 1918-2024. (Middle) Future H1N1 sequon locations predicted by FSF using a mutation frequency calculation based on a random selection of ≤ 500 H1N1 HA sequences per year for all years from 2009-2024. (Bottom) Future H3N2 sequon locations predicted by FSF using a mutation frequency calculation based on a random selection of ≤ 500 H3N2 HA sequences per year for all years from 1968-2024.

H1N1 All Years

| Overall ranking | Near sequon site | Near-sequon frequency | Average sidechain exposure | Mutation rate | Gamma category | Gamma certianty | Near-sequon designation | Average NS ranking | Average VNS ranking | Average UNS ranking |
|---|---|---|---|---|---|---|---|---|---|---|
| 1 | 286 | 0.1 | 64.50% | 4.6243 | 8 | 0.999991929 | Very near-sequon | 7 | 1 | NA |
| 2 | 111 | 0.1 | 85.10% | 4.5318 | 8 | 0.961737947 | Very near-sequon | 8 | 1 | NA |
| 3 | 111 | 0.1 | 78.60% | 1.3607 | 6 | 0.744258253 | Very near-sequon | 11 | 1 | NA |
| 4 | 201 | 1 | 51.18% | 1.3296 | 6 | 0.711083036 | Ultra near-sequon | 11.6 | 1.3 | 1 |
| 5 | 136 | 1 | 31.66% | 1.0194 | 5 | 0.510145247 | Very near-sequon | 12.6 | 2.3 | NA |
| 6 | 245 | 1 | 21.52% | 0.9383 | 5 | 0.465989051 | Very near-sequon | 14.6 | 3.3 | NA |
| 7 | 256 | 1 | 65.71% | 0.6547 | 4 | 0.471340697 | Very near-sequon | 15.6 | 4.3 | NA |
| 8 | 181 | 0.1 | 37.70% | 0.5096 | 4 | 0.544519551 | Ultra near-sequon | 18 | 6 | 2 |
| 9 | 73 | 1 | 26.31% | 0.25 | 3 | 0.42380528 | Very near-sequon | 21.5 | 5.4 | NA |

H1N1 2009+

| Overall ranking | Near sequon site | Near-sequon frequency | Average sidechain exposure | Mutation rate | Gamma category | Gamma certianty | Near-sequon designation | Average NS ranking | Average VNS ranking | Average UNS ranking |
|---|---|---|---|---|---|---|---|---|---|---|
| 1 | 286 | 0.1 | 64.50% | 4.3548 | 8 | 0.999946882 | Very near-sequon | 4 | 1 | NA |
| 2 | 111 | 0.1 | 85.10% | 2.3362 | 7 | 0.832885697 | Very near-sequon | 10 | 1 | NA |
| 3 | 136 | 1 | 31.90% | 1.5147 | 6 | 0.526586082 | Very near-sequon | 11.2 | 1.2 | NA |
| 4 | 201 | 1 | 51.14% | 1.3234 | 6 | 0.648165763 | Ultra near-sequon | 12.2 | 2.2 | 1 |
| 5 | 256 | 1 | 65.72% | 1.0935 | 6 | 0.494117309 | Very near-sequon | 14.2 | 3.2 | NA |
| 6 | 111 | 0.1 | 78.60% | 1.0706 | 6 | 0.526596067 | Very near-sequon | 15 | 4 | NA |
| 7 | 181 | 0.1 | 37.70% | 0.6317 | 4 | 0.434216146 | Ultra near-sequon | 16 | 5 | 2 |
| 8 | 245 | 1 | 21.47% | 0.6194 | 4 | 0.362708452 | Very near-sequon | 15.7 | 4.4 | NA |
| 9 | 73 | 1 | 26.38% | 0.1835 | 2 | 0.422196462 | Very near-sequon | 21.5 | 5.4 | NA |

H3N2 All Years

| Overall ranking | Near sequon site | Near-sequon frequency | Average sidechain exposure | Mutation rate | Gamma category | Gamma certianty | Near-sequon designation | Average NS ranking | Average VNS ranking | Average UNS ranking |
|---|---|---|---|---|---|---|---|---|---|---|
| 1 | 160 | 1 | 84.43% | 5.2486 | 8 | 0.999979796 | Ultra near-sequon | 2 | 1 | 1 |
| 2 | 138 | 0.5 | 67.26% | 5.1587 | 8 | 0.969068272 | Very near-sequon | 4 | 2 | NA |
| 3 | 174 | 1 | 79.49% | 3.4833 | 7 | 0.607051602 | Very near-sequon | 5.5 | 2.5 | NA |
| 4 | 223 | 0.9 | 63.66% | 2.3371 | 7 | 0.992646267 | Very near-sequon | 7.6 | 3.6 | NA |
| 5 | 206 | 1 | 44.09% | 2.3287 | 7 | 0.987549431 | Very near-sequon | 8.5 | 4.4 | NA |
| 6 | 213 | 0.1 | 44.10% | 2.2362 | 7 | 0.891834972 | Very near-sequon | 10 | 5 | NA |
| 7 | 280 | 1 | 94.70% | 2.1309 | 7 | 0.783493861 | Very near-sequon | 11.5 | 5.5 | NA |
| 8 | 121 | 1 | 40.17% | 1.4466 | 6 | 0.679889132 | Very near-sequon | 12.5 | 6.5 | NA |
| 9 | 140 | 0.2 | 100.00% | 1.2228 | 6 | 0.693254615 | Ultra near-sequon | 13 | 6.5 | 2 |
| 10 | 69 | 0.9 | 42.13% | 0.7351 | 5 | 0.381898704 | Very near-sequon | 15.7 | 7.8 | NA |
| 11 | 175 | 1 | 88.49% | 0.4438 | 4 | 0.452966492 | Very near-sequon | 18.5 | 8.6 | NA |
| 12 | 168 | 0.6 | 16.97% | 0.0618 | 1 | 0.573958363 | Very near-sequon | 20.8 | 9.7 | NA |

To explain the lack of N-linked glycosylation in unexposed regions, we propose that SAEs with unexposed asparagine residues either remain unglycosylated or become glycosylated before protein folding, introducing detrimental conformational changes. Given

the co-translational nature of glycan addition via oligosaccharyltransferase, a region that is unexposed post-folding may be exposed at the time of glycosylation. Adding a bulky oligosaccharide moiety to such a site, however, would likely result in significant conformational changes during folding, rendering the HA protein ineffective. If the SAE site folds correctly into its unexposed state before glycan binding, glycosylation will not occur. As a result, the mutation will provide no selective advantage via epitope shielding and is less likely to rise to prominence. Sequons may still emerge in these locations, but their immediate biological relevance would be less significant given that they either wouldn't host an N-linked glycan, or the new glycan would introduce detrimental conformational changes.

In conjunction with the solvent accessibility results, the mutation frequency analysis proved a powerful predictor of which near-sequons would become true sequons in the human IFV dataset. In particular, ranking very near-sequons based on the mutation frequencies of their mismatched AAs resulted in almost all correct near-sequons being found at the top of their respective lists. Of the 25 correctly predicted very near-sequons, 24 were ranked in the top 10 very near-sequons, 19 in the top 5, and 17 were ranked in the top 3 based on site-specific mutation frequency. It is also worth noting that all correctly predicted near-sequons with one exception had associated site-specific mutation rates exceeding 1.0, meaning the rates of the sites that would ultimately mutate and generate sequons were almost always greater than the average AA mutation rate across the HA. Interestingly, the mutation frequency analysis proved effective even when calculated from a minimal number of preexisting sequences, especially in the case of H1N1 viruses in the early to mid-20th century. H1N1 mutation analyses for 1934 and 1935 successfully placed all correct very near-sequons in the top 7, despite the calculation being based on only 59 and 64 preexisting H1N1 strains, respectively. It is unclear if the abundance of available strains from the early 2000s and onwards significantly improves the power of mutation frequency analysis as a predictive tool, as there have been few SAEs in H1N1 and H3N2 since the turn of the century. While it seems reasonable that larger sequence datasets could improve the predictive power of mutation analyses, more data is needed.

There are several limitations to the FSF method worth addressing. FSF excels at predicting SAE locations when the analyzed strain is closely related to the SAE strain, and the SAE occurs in a region with a high mutation rate. It isn't guaranteed that next year's predominant IFVs will be most closely related to this year's, particularly in the case of H1N1. If a historic strain distantly related to circulating strains were to reemerge, FSF analyses using circulating strains would be less accurate. This is also true in the event of novel strain emergence from animal reservoirs. Additionally, FSF only predicts the emergence of sequons and does not predict whether those sequons will become glycosylated. The sidechain exposure analysis used by FSF may help provide some insight into this, as unexposed sequons will seldom be glycosylated. There are, however, other factors influencing whether a glycan will bind to a sequon. There are, however, other factors influencing whether a glycan will bind to a sequon. The local secondary structure, the hydrophobicity of the region, and the conformational flexibility of the region have all been shown to influence sequon occupancy in a structural analysis of glycoproteins[39]. Additional factors include the identity and charge of amino acids flanking the tripeptide sequon[40], as well as the sequon's proximity to the C-terminus[41]. Fortunately, there are numerous existing programs designed to determine which sequons in a protein are likely to become glycosylated [42–46]. Running the most probable near-sequon mutants identified by FSF through these programs may further enhance glycosylation site predictions.

The analysis of currently circulating H1N1 and H3N2 IFV strains revealed sites that may become glycosylated in the future, assuming future SAE strains are most closely related to

those circulating presently. In the historical analysis of human IFVs, every correct near-sequon prediction was tagged as a very near-sequon. Consequently, only very near-sequons (or ultra near-sequons) were considered serious candidates for SAEs in the near future. All near-sequons appearing in the lists were the highest-ranking sites predicted by FSF. It is important to track these near-sequons over time as a near-sequon present in strains this year may not persist in strains next year, and vice versa. For example, position 142 in the H1N1 HA is not included in the list of predicted sites given that it is not a single nucleotide change away from becoming a sequon, but the mismatched AA in this near-sequon (position 144) currently has the highest mutation rate in the entire H1 domain of the H1N1 HA protein and may become a sequon in time. For this reason, an analysis of presently circulating IFVs should be conducted every few years to catch the new very near-sequons that emerge. That being said, should a new N-linked glycosylation sequon emerge soon in either H1N1 or H3N2, it is likely included in Table 5.

The historic analysis of IFVs with FSF demonstrated that it is often possible to predict N-linked glycosylation sequon locations before they emerge in H1N1 and H3N2 HA proteins. The predictive capacity also seems to extend to swine H1N1 viruses, though the mutation frequency analysis was less effective. Whether this predictive capacity extends to HA proteins in other IFV types, such as influenza B viruses or those circulating in avian reservoirs, should be tested. Glycosylation changes on the IFV neuraminidase surface protein are also likely to be predictable. This predictive capacity may even extend to other viral surface proteins such as the heavily glycosylated envelope glycoprotein of the human immunodeficiency virus (HIV), which is known to alter glycosylation patterns to take advantage of glycan shielding [47], or the SARS-CoV-2 spike protein which has also been shown to evade antibodies via an N-linked glycan shield that protects approximately 40% of the underlying protein surface from antibody recognition [48]. FSF serves as proof of concept, and the predictive power of sequon prediction programs like it can undoubtedly be improved. With a larger dataset of recorded SAEs in viral surface proteins, machine learning could be implemented to reveal other useful parameters for sequon prediction or assign weights to existing parameters to enhance prediction accuracy. One promising parameter not explored in this study is the frequency of near-sequons in IFVs circulating worldwide at a given time. Near-sequon frequencies were reported in Table 5, though it is unclear how these may impact the likelihood of a given sequon emerging. Once refined, sequon prediction programs like FSF could work in synergy with existing algorithms. This integration would enhance antigenic escape modeling and guide the development of effective vaccines and monoclonal antibody therapies.

## Supporting information

**S1 Appendix. List of tested HA sequences.**
(ZIP)

**S1 File. Future Sequon Finder source code.**
(ZIP)

**S2 File. Influenza HA-based phylogenetic trees.**
(ZIP)

**S3 File. Influenza HA protein structure files.**
(ZIP)

## Acknowledgments

The authors would like to thank Dr. Daven Presgraves and Dr. Christopher Anderson of the University of Rochester for their helpful suggestions regarding phylogenetic analysis techniques and influenza sequence data organization, respectively.

## Author contributions

**Conceptualization:** Shane P. Bryan, Martin S. Zand.

**Data curation:** Shane P. Bryan.

**Formal analysis:** Shane P. Bryan.

**Investigation:** Shane P. Bryan.

**Methodology:** Shane P. Bryan, Martin S. Zand.

**Project administration:** Martin S. Zand.

**Resources:** Martin S. Zand.

**Software:** Shane P. Bryan.

**Validation:** Shane P. Bryan.

**Visualization:** Shane P. Bryan.

**Writing – original draft:** Shane P. Bryan.

**Writing – review & editing:** Shane P. Bryan, Martin S. Zand.

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
