## [Decision Letter · Decision Letter 0]

12 May 2025

PONE-D-25-18727Future Sequon Finder - A novel approach for predicting future N-linked glycosylation sequon locations on viral surface proteinsPLOS ONE

Dear Dr. Zand,

Thank you for submitting your manuscript to PLOS ONE. After careful consideration, we feel that it has merit but does not fully meet PLOS ONE’s publication criteria as it currently stands. Therefore, we invite you to submit a revised version of the manuscript that addresses the points raised during the review process.

We look forward to receiving your revised manuscript.

Kind regards,

Cheorl-Ho Kim, Ph.D.

Academic Editor

PLOS ONE

Journal Requirements:

“The project described in this publication was supported by the University of Rochester CTSA award number UL1 TR002001 from the National Center for Advancing Translational Sciences (MZ), and award number R01 AI 134058 from the National Institute for Immunology, Allergy, and Infectious Diseases from the National Institutes of Health (MZ, SB) . The content is solely the responsibility of the authors and does not necessarily represent the official views of the National Institutes of Health.”

Additional Editor Comments :

Dear Dr Zand,

Thank you for your submission of your interesting study to Plos One.

I have completed the review process and am pleased to inform that your manuscript can be accepted for publication after your minor revision.

As you read, you can easily and simply revise the original manuscript.

Thank you for your submission.

Looking forward to receiving your revision.

Cheorl-Ho Kim, Ph.D

Editor

Reviewers' comments:

Reviewer's Responses to Questions

**Comments to the Author**

1. Is the manuscript technically sound, and do the data support the conclusions?

Reviewer #1: Yes

Reviewer #2: Yes

2. Has the statistical analysis been performed appropriately and rigorously? 

Reviewer #1: Yes

Reviewer #2: Yes

3. Have the authors made all data underlying the findings in their manuscript fully available?

Reviewer #1: Yes

Reviewer #2: Yes

4. Is the manuscript presented in an intelligible fashion and written in standard English?

Reviewer #1: Yes

Reviewer #2: Yes

5. Review Comments to the Author

Reviewer #1: (1) The authors mainly consider the exposure of newly acquired N-glycosylation sites on the molecular surface as an indicator of functional relevance. However, it is also important to note that the formation of the HA trimer is essential for glycan-binding activity. Since trimerization is distinct from immune surveillance mechanisms, it may impose constraints on further molecular evolution. This aspect should be discussed in the Discussion section of the manuscript.

(2) It is generally considered that N-linked glycosylation enables viruses to evade immune surveillance mechanisms. Has the efficacy of this mechanism been evaluated using other viruses? And if there are issues to be considered in the program, what aspects should be taken into account? On this point, the authors shold include their opinions in the main text, which would provide valuable information for readers conducting similar studies.

Reviewer #2: This manuscript by Zand et al. titled describes a program (Future Sequon Finder) to predict where N-linked glycosyltion sites will likely emerge on viral surface proteins, tested primarily on influenza hemagglutinin. This paper addresses a relevant and timely problem, but several concerns need to be addressed before it can be accepted for publication.

- The program uses genetic data; however, the method by which the DNA sequence is translated into a protein sequence remains unclear. There are six open reading frames, and it is uncertain how the program will determine the correct open reading frame.

- The proposed program predicts the likely location for glycosylation; however, it does not guarantee that this site will indeed undergo glycosylation. While the exposure of the side chain increases chances of potential glycan attachment, other factors also influence whether glycosylation will occur or not. Can authors comment on that?

- Author did not consider oligomerization of protein in when calculation exposed potential glycosylation sites. Could it be a limitation of the program.

- The definition of a sequon should be clearly stated. Given that not all glycosylation sites conform to the Asn-X-Ser/Thr rule, the specific amino acid sequences the program looks for are required.

- Authors should provide a access to the program if possible.

6. PLOS authors have the option to publish the peer review history of their article (what does this mean?). If published, this will include your full peer review and any attached files.

Reviewer #1: No

Reviewer #2: **Yes: **Sushil Mishra

---

## [Author Response · Author response to Decision Letter 1]

25 Jun 2025

We have made the requested corrections, specifically:

and

We have changed the file names of our supplemental figures and we have updated the formatting of the author affiliations to be in accordance with PLOS One’s formatting guidelines.

In accordance with PLOS ONE's guidelines, we have created a publicly accessible GitHub repository containing all author-generated code used in this study. To ensure long-term accessibility, we have archived the repository with Zenodo and generated a DOI. The DOI has been cited in the manuscript.

“The project described in this publication was supported by the University of Rochester CTSA award number UL1 TR002001 from the National Center for Advancing Translational Sciences (MZ), and award number R01 AI 134058 from the National Institute for Immunology, Allergy, and Infectious Diseases from the National Institutes of Health (MZ, SB) . The content is solely the responsibility of the authors and does not necessarily represent the official views of the National Institutes of Health.”

This statement has been added.

The captions for our Supporting Information files have been adjusted to match PLOS One’s formatting requirements. Optional figure legends have been omitted due to the self-explanatory nature of the file names.

We have removed several references that are no longer cited directly in the text, or are better represented by another reference already in the bibliography.

The titles of the removed references are as follows:

1.

Cited manuscript:

Origin of the pandemic 1957 H2 influenza A virus and the persistence of its possible progenitors in the avian reservoir (Schafer)

Reason removed:

The section containing the reference was removed.

2.

Cited manuscript:

The re-emergence of H1N1 influenza virus in 1977: a cautionary tale for estimating divergence times using biologically unrealistic sampling dates (Wertheim)

Reason removed:

The section containing the reference was removed.

3.

Cited manuscript:

The Reemergent 1977 H1N1 Strain and the Gain-of-Function Debate (Rozo)

Reason removed:

The section containing the reference was removed.

4.

Cited manuscript:

Influenza Research Database: An integrated bioinformatics resource for influenza virus research (Zhang)

Reason removed:

This is a reference to the Influenza Research Database, which had joined with the BV-BRC prior to writing this manuscript. Given that the data used in this study was obtained from the BV-BRC, all references to the IRD were replaced with references to the BV-BRC.

Additionally, several references were added to support new text in the revised manuscript.

The titles of the added references are as follows:

1.

Statistical analysis of the protein environment of N-glycosylation sites: implications for occupancy, structure, and folding (Petrescu)

2.

N-glycosylation efficiency is determined by the distance to the C-terminus and the amino acid preceding an Asn-Ser-Thr sequon (Baño-Polo)

Review Comments to the Author

Reviewer #1: (1) The authors mainly consider the exposure of newly acquired N-glycosylation sites on the molecular surface as an indicator of functional relevance. However, it is also important to note that the formation of the HA trimer is essential for glycan-binding activity. Since trimerization is distinct from immune surveillance mechanisms, it may impose constraints on further molecular evolution. This aspect should be discussed in the Discussion section of the manuscript.

Several sentences have been added in the manuscript to address the trimerization of the HA.

(2) It is generally considered that N-linked glycosylation enables viruses to evade immune surveillance mechanisms. Has the efficacy of this mechanism been evaluated using other viruses? And if there are issues to be considered in the program, what aspects should be taken into account? On this point, the authors should include their opinions in the main text, which would provide valuable information for readers conducting similar studies.

The final paragraph of the manuscript has been modified to address the first point by elaborating on the extent to which spike proteins are glycosylated in SARS-CoV-2, and the third-to-last paragraph of the manuscript has been modified to address the second point by elaborating on the weaknesses of the current approach.

Reviewer #2: This manuscript by Zand et al. titled describes a program (Future Sequon Finder) to predict where N-linked glycosyltion sites will likely emerge on viral surface proteins, tested primarily on influenza hemagglutinin. This paper addresses a relevant and timely problem, but several concerns need to be addressed before it can be accepted for publication.

- The program uses genetic data; however, the method by which the DNA sequence is translated into a protein sequence remains unclear. There are six open reading frames, and it is uncertain how the program will determine the correct open reading frame.

The following sentences have been added to the manuscript: “Sequences were curated to ensure they began with the correct start codon, eliminating reading frame ambiguity for translation. Sequences that did not begin with a start codon were eliminated.” This curation was performed by “ifvDataCleaner”, a small helper program found in the “additional_tools” folder of the program files.

- The proposed program predicts the likely location for glycosylation; however, it does not guarantee that this site will indeed undergo glycosylation. While the exposure of the side chain increases chances of potential glycan attachment, other factors also influence whether glycosylation will occur or not. Can authors comment on that?

Several sentences describing other factors that contribute to sequon occupancy have been added to the discussion section of the manuscript.

- Author did not consider oligomerization of protein in when calculation exposed potential glycosylation sites. Could it be a limitation of the program.

Several sentences have been added in the manuscript to address the trimerization of the HA.

- The definition of a sequon should be clearly stated. Given that not all glycosylation sites conform to the Asn-X-Ser/Thr rule, the specific amino acid sequences the program looks for are required.

- Authors should provide a access to the program if possible.

This has been done. The program is now accessible through GitHub and archived through Zenodo.

Sincerely,

Martin S. Zand MD PhD

---

## [Editor Report · Decision Letter 1]

27 Jun 2025

Future Sequon Finder - A novel approach for predicting future N-linked glycosylation sequon locations on viral surface proteins

PONE-D-25-18727R1

Dear Dr. Zand,

We’re pleased to inform you that your manuscript has been judged scientifically suitable for publication and will be formally accepted for publication once it meets all outstanding technical requirements.

Kind regards,

Cheorl-Ho Kim, Ph.D.

Academic Editor

PLOS ONE

Additional Editor Comments (optional):

Dear Dr Zand,

Thank you for your submission and revision.

I would like to accept your revision, as I have previously requested to revise.

Basically, it should be published and I am pleased to deal with your study.

Thanks a lot

Cheorl-Ho Kim Ph.D

Editor
---

## [Editor Report · Acceptance letter]

PONE-D-25-18727R1

PLOS ONE

Dear Dr. Zand,

I'm pleased to inform you that your manuscript has been deemed suitable for publication in PLOS ONE. Congratulations! Your manuscript is now being handed over to our production team.

Kind regards,

on behalf of

Professor Cheorl-Ho Kim

Academic Editor

PLOS ONE